# DATASET TRANSFORMATIONS TRADE-OFFS TO ADAPT MACHINE LEARNING METHODS ACROSS DOMAINS

## ABSTRACT

Machine learning-based methods have been proved to be quite successful in different domains. However, applying the same methods across domains is not a trivial task. In the literature, the most common approach is to convert a dataset into the same format as the original domain to employ the same architecture that was successful in the original domain. Although this approach is fast and convenient, we argue it is suboptimal due to the lack of tailoring to the specific problem at hand. To prove our point, (1) we exhaustively examine the dataset transformations that are used in the literature to adapt machine learning-based methods across domains; (2) we show that these dataset transformations are not always beneficial; and (3) we show the drawbacks of converting the dataset to adapt a Machine learning-based method to a different domain. To quantify how different the original dataset is with respect to the transformed one, we compute the dataset distances via Optimal Transport. Also, we present simulations with the original and transformed data to show that the data conversion is not always needed or beneficial.

## 1 INTRODUCTION

Machine learning (ML) methods are revolutionizing and transforming a wide range of disciplines. This is being boosted by unprecedented integration of new real-time monitoring, sensing, control, and communication devices, which give us a wealth of data to propel the ML techniques. However, ML research is usually done in computer science to solve problems in computer vision (CV), audio recognition, among others. The same techniques used in these domains are adapted to many other different fields and areas. For example, many of these techniques are used to monitor, control, and operate cyber-physical systems (CPSs) such as tracking object systems, water distribution systems, power systems, among others. Therefore, nowadays, data-driven monitoring and control play an important role for Cyber-Physical Systems (CPSs), e.g., smart grids, autonomous automobile systems, robotic systems, etc. Jazdi (2014). For example, Duchesne et al. (2020) gives an overview of ML methods for reliability and management in energy systems. Most of such applied methods fall into classification and regression problems using supervised and unsupervised learning. For example, Marot et al. (2018) automatically segments large-scale power grids into coherent zones for the management of the grid for control operators with ML techniques. However, little research has been done in maximizing the adaptation of ML techniques across domains in a unified framework.

In this work, we claim that while adopting ML techniques solves specific problems, the way to adapt these techniques also open the door on CPSs vulnerability with dire consequences Alguliyev et al. (2018). For example, one can intentionally distort inputs according to how one applies ML method to cause an incorrect control action due to an estimation mistake. Known as adversarial attacks Kurakin et al. (2018), many studies show that outputs from classifiers can be easily changed with imperceptible changes to the input in the computer science domain Carlini & Wagner (2017); Szegedy et al. (2013). In the CPS context, this is known as a False Data Injection Attack (FDIA), where an attacker intercepts and maliciously changes the system measurements to cause harm in the real world. For instance, a cyber-attack in a power system could cause a system operator to take wrong control actions causing a blackout. Similarly, a cyber-attack could cause fatal autopilot crashes in autonomous automobile systems Banks et al. (2018); Dikmen & Burns (2017).

Adapting the ML methods from the originally proposed area to a new one (CV to CPSs, for example) is a non-trivial task that has not been thoroughly analyzed in the literature. The general consensus in the ML community says that there are no standard recipes to apply a given method across multiple domains. The dataset examples are different across domains; for example, whereas the CV dataset samples are images that are represented as three-dimensional tensors, in CPSs the dataset samples are one-dimensional tensor measurements from multiple devices with a timestamp. Then, in the literature, there are two main approaches to adapt methods to a specific domain. (1) Modify the ML learning architecture to fit the dataset samples format. For example, if the original domain is CV in which a convolutional neural network (CNN) in employed, the adapted ML technique will use as a base a fully connected neural network to handle the one-dimensional data in the targeted domain. (2) Transform the new dataset samples into the same format as the original domain to use the same ML architecture. For example, if in the original domain a CNN with a three-dimensional as input, and the targeted domain is a CPS with one-dimensional tensors, then the one-dimensional is transformed into a grid-like format to use a CNN as in the original domain. This begs the question, What are the benefits and downsides of using the first or second approach to adapt ML techniques across different domains?

To better adapt ML to different domains such as CPS systems, it is important to understand the benefits and possible downsides for different methods using particular architectures and dataset types. In this paper, we explore the benefits and downsides of using different approaches to adapt ML methods across disciplines. There are multiple ML methods. To simplify our analysis, we choose to study classifiers. This will allow us to quantify the benefits of the classifier's accuracy. To quantify or evaluate the downsides, we analyze how different datasets and ML architecture are impacted by adversarial attacks. We will show the trade-offs of using different dataset formats and model architectures. In specific, this paper demonstrates that while CNN improves model accuracy, they are more sensitive to adversarial examples than fully connected neural networks (FCNN).

## 2 RELATED WORK

**Computer vision** (CV) is an area that studies how computers can gain high-level understanding from digital images or videos, which utilizes 2- and 3-dimensional data with multi-channels. Convolutional neural networks (CNN) are the most important architecture for CV and can be 2- and 3-dimensional (CNN with higher dimensions are possible but rarely used). These CNN are designed to learn filters from data where there is translation invariant. CNN can pull out this translational invariance that images inherently have. For example, if you want to recognize a cat in an image, it should not matter the cat's location in the picture. Computer vision also deals with 3-dimensional images, for example, CT and MRI scans. To exploit these 3-dimensional data, 3-d CNN are employed.

**Adversarial attacks**. Deep Convolutional Neural Networks (CNNs) models are highly susceptible to adversarial examples Goodfellow et al. (2014); Moosavi-Dezfooli et al. (2017); Szegedy et al. (2013). Research on this topic has been done mainly for image classifiers and face recognition systems. These adversarial attacks or adversarial samples have drawn the community's attention; in 2018, there was an *Adversarial Attacks and Defences Competition* Kurakin et al. (2018).

A lot of work to create adversarial samples has been done, including techniques such as the L-BFGS attack Szegedy et al. (2013), FGSM Goodfellow et al. (2014), and the CW attack Carlini & Wagner (2016). In specific for classifiers in the white-box context, in addition to the model knowledge, the attacker knows the set of classes $Y$ and the set of valid inputs $X$ to the classifier. These methods do not explicitly make use of the latent features to create adversarial examples. For example, the conventional way to create these adversarial examples is by maximizing the log-likelihood of the target class $y_t$ over a $\epsilon$-radius ball around the original input (which is usually represented as a vector of $d$ pixels in the range $[0, 1]$) Athalye et al. (2017):

$$\underset{x'}{\arg\max} \log P\left(y_t | x'\right)$$
$$\text{subject to } \left\|x' - x\right\|_p < \epsilon \tag{1}$$
$$x' \in [0, 1]^d.$$

**False data injection in power systems**. Past work in the power systems area shows the vulnerability of the state estimation functionality. It is based on the observability of the system state.

Some works show different attacking approaches Mohammadpourfard et al. (2017). Based on such analysis, the impact of an attack on the system is also evaluated Mohammadpourfard et al. (2019). Mathematically, the FDIA can be expressed as:

$$\min_{\mathbf{m}_a} \left\| \mathbf{m}_a - h\left(\hat{\mathbf{v}}_{st}\right) \right\|_2^2$$

$$\text{subject to } \mathbf{m}_a \leq \mathbf{m} + \mathbf{a},$$

(2)

where $\mathbf{m}$ and $\mathbf{m}_a$ are the set of original and corrupted measurements, respectively; $\hat{\mathbf{v}}_{st}$ is a vector of the state variables, $h$ is the state estimator non-linear model, and $\mathbf{a}$ is the attack vector to alter the original set of measurements.

## 3 BACKGROUND

This section presents the Optimal Transport (OT) problem that will allow us to quantify how different similar two distributions or datasets are from each other.

### 3.1 BACKGROUND ON OPTIMAL TRANSPORT

Optimal transport (OT) is compares probability distributions (Villani, 2003; 2008). OT leverages the geometry of the underlying space, making them ideal for comparing distributions, shapes and point clouds (Peyré & Cuturi, 2019).

The OT problem considers a complete and separable metric space $\mathcal{X}$, along with probability measures

$$\alpha \in \mathcal{P}(\mathcal{X}), \beta \in \mathcal{P}(\mathcal{X}).$$

(3)

The Kantorovich formulation Kantorovitch (1942) of the transportation problem can be writeen as:

$$\text{OT}(\alpha, \beta) \triangleq \min_{\pi \in \Pi(\alpha, \beta)} \int_{\mathcal{X} \times \mathcal{X}} c(x, y) \, \mathrm{d}\pi(x, y),$$

(4)

where $c(\cdot, \cdot) : \mathcal{X} \times \mathcal{X} \to \mathbb{R}^+$ is a cost function (the "ground" cost), and the set of couplings $\Pi(\alpha, \beta)$ consists of joint probability distributions over the product space $\mathcal{X} \times \mathcal{X}$ with marginals $\alpha$ and $\beta$, that is,

$$\Pi(\alpha, \beta) \triangleq \{\pi \in \mathcal{P}(\mathcal{X} \times \mathcal{X}) \mid P_{1\#}\pi = \alpha, P_{2\#}\pi = \beta\}.$$

(5)

Whenever $\mathcal{X}$ is equipped with a metric $d_{\mathcal{X}}$, it is natural to use it as ground cost, e.g.,

$$c(x, y) = d_{\mathcal{X}}(x, y)^p$$

(6)

for some $p \geq 1$. In such case,

$$\mathbf{W}_p(\alpha, \beta) \triangleq \text{OT}(\alpha, \beta)^{1/p}$$

(7)

is called the $p$-Wasserstein distance. The case $p = 1$ is also known as the Earth Mover's Distance (Rubner et al., 2000).

The measures $\alpha$ and $\beta$ are rarely known in practice. Instead, one has access to finite samples

$$\{\mathbf{x}^{(i)}\} \in \mathcal{X}, \{\mathbf{y}^{(j)}\} \in \mathcal{X}.$$

(8)

In that case, one can construct discrete measures

$$\alpha = \sum_{i=1}^{n} \mathbf{a}_i \delta_{\mathbf{x}^{(i)}}$$

(9)

and

$$\beta = \sum_{i=1}^{m} \mathbf{b}_i \delta_{\mathbf{y}^{(j)}},$$

(10)

where $\mathbf{a}$, $\mathbf{b}$ are vectors in the probability simplex, and the pairwise costs can be compactly represented as an $n \times m$ matrix $\mathbf{C}$, i.e.,

$$\mathbf{C}_{ij} = c(\mathbf{x}^{(i)}, \mathbf{y}^{(j)}). \tag{11}$$

In this case, Equation equation 4 becomes a linear program. Adding an entropy regularization, namely

$$\mathrm{OT}_\epsilon(\alpha, \beta) \triangleq \min_{\pi \in \Pi(\alpha, \beta)} \int_{\mathcal{X}^2} c(x, y) \, \mathrm{d}\pi(x, y) + \epsilon \mathbf{H}(\pi \, | \, \alpha \otimes \beta), \tag{12}$$

where

$$\mathbf{H}(\pi \, | \, \alpha \otimes \beta) = \int \log(\mathrm{d}\pi / \, \mathrm{d}\alpha \, \mathrm{d}\beta) \, \mathrm{d}\pi \tag{13}$$

is the relative entropy that results into a problem that can be solved more efficiently (Cuturi, 2013; Altschuler et al., 2017) and with better sample complexity (Genevay et al., 2019) than the original one.

We will use OT to measure the distance between datasets as was done by Alvarez-Melis & Fusi (2020), see Fig. 1. In specific, we'll measure the distance between regular and adversarial datasets.

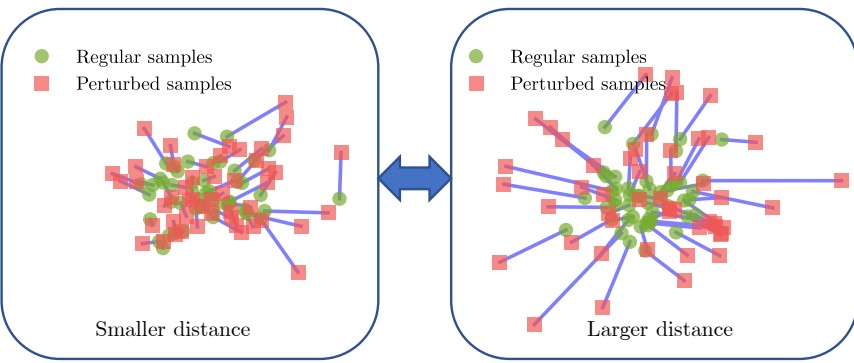

Figure 1: Illustration of OT for regular and perturbed datasets.

## 4 TRADE-OFFS OF DATASET TRANSFORMATIONS

Machine learning (ML) approaches have been extensively applied on different domains to solve unsupervised and supervised tasks Olowononi et al. (2020); Duchesne et al. (2020). Unsupervised tasks mainly encompassed clustering, and supervised tasks primarily included classification problems. In the last section, the different types of data and the most NN architectures were described. Choosing a specific NN design will give you a *prior* or impose a bias on the model. For example, if you use a vanilla MLP NN, each input will interact with each output. This means either you assume that all the input data is correlated or have no idea of a structure that you might exploit. If you use a CNN, you are assuming that your grid-like data are locally correlated but not further data Xu et al. (2018b); Jeong et al. (2017); Xu et al. (2017). In this way, you are imposing a bias or a prior over the model, which in this case, will be reflected in the learned kernels by the CNN. On the other hand, if your data comes in sequence, you can choose to use a RNN that will exploit this prior knowledge of the problem.

ML techniques have been used across different domains. In CPSs on power systems to do classification or regression, for example. In classification for stability, the work Alimi et al. (2020) gives an overview of the work in this area. The takeaway is that the applied methods are in supervised ML techniques to classify an event. For example, the work Wang et al. (2016) assesses the power system stability after the occurrence of a disturbance. They use a feedforward NN. For regression, ML methods have been applied to forecast. For example, the work Xu et al. (2018a) tries to predict power load with a deep-belief network.

Chen et al. (2018) used time-series data such as wind and solar generation with 5- and 60-minute resolution. They separate the data by day and then stack the time series into a matrix. So, for hourly data, they form a $24 \times 24$ matrix. Then a convolutional neural network is employed. Gupta et al. (2018) monitored three time-series with each PMU at every generator unit: voltage magnitude, voltage angle, and rate of change of voltage angle. The time series of all generators are stacked to form a 2-D matrix for each of the three variables. This results in three 2-D matrices. Then, these three images are combined to form a three-channel, 2-D image, which they call a heatmap. Finally, a convolutional neural network is trained on the heat maps to classify the power system state. Cai & Hill (2022) take voltage (V), active (P), and reactive (Q) power measurements at each bus. With each V, P, and Q, a matrix is formed. Then the three matrices V, P, and Q are combined to create an "RGB" image. Shi et al. (2020) converted the data as follows. Given $N$ buses PMU phasor voltage time series with length $T$, two real matrices $N \times T$ are formed to get a two-channel image. Then, this is the input to the convolutional neural network. Ren et al. (2020) converted the frequency time-series to a matrix with wavelet decomposition and polar coordinate Gramian Angular Field. Then, this grid-like data can be used as input to the convolutional neural network. All these applications of ML methods use a given model architecture and dataset format without analyzing the implications of different alternatives.

In summary, we can list the main ideas or principles by which data is transformed:

1. Single time-series

    (i) The most straightforward and least principled way to convert one-dimensional data into a grid-like (or image-like) format is to resize your time-series with length $T$ to a matrix with size $n \times m = T$, as Wang et al. (2017) did.

    (ii) A more principled way to convert one-dimensional time-series data into a grid-like format is to resize your time-series, exploiting the features of your signal. The time series can be segmented in chunks of one day, and then these chunks can be stacked into a matrix. Chen et al. (2018) followed this approach. For hourly data, for example, they formed a $24 \times 24$ matrix.

    (iii) Given a one-dimensional time-series, a transform can be applied to every sample to obtain grid-like data from the time series. In speech recognition, they use the Gabor transform to obtain a spectrogram, for example. In power systems, Ren et al. (2020) converted the frequency time-series to a matrix with wavelet decomposition and polar coordinate Gramian Angular Field.

2. Multiple time-series

    (i) When multiple time-series are available, the data conversion is carried out in a different way. A set of time series of the same variable from many sources can be stacked into a matrix. This process can be applied to every variable resulting in the same number of matrices ($c$). Then, these formed matrices can be stacked in a tensor to form an "image" with $c$ channels. Examples of these approach are Gupta et al. (2018); Cai & Hill (2022); Shi et al. (2020).

### 4.1 DIFFERENT DATASET FORMATS

Given all the dataset transformations previously described, in this part we introduce a classification for different dataset types. The use of each dataset type will dictate which neural network (NN) architecture to use. Starting with the most simple dataset format, let $\mathcal{D}_{T_1}$ be a dataset type one defined as

$$\mathcal{D}_{T_1} = \left\{ \left( x_{m,b}^{(n)}, y^{(n)} \right) : m = [\![M]\!], b = [\![B]\!] \right\}_{n=1}^{N}, \tag{14}$$

where $x_{m,b}^{(n)} \in \mathbb{R}^T$ is the $n$ time-series sample of variable $m$ at location $b$ with length $T$, $M$ is the number of different measured variables, $[\![M]\!] = \{1, \ldots, M\}$, $B$ is the number of different locations, $[\![B]\!] = \{1, \ldots, B\}$, $N$ is the number of samples in the dataset, and $y^{(n)}$ is the label associated with the $n$ sample.

As shown before, in the literature, they convert a vector into a matrix. To do so, we can stack the $B$ samples $x_{m,b}^{(n)}$ measuring the same variable from different locations as

$$V_m^{(n)} = \begin{bmatrix} x_{m,1}^{(n)} \\ \vdots \\ x_{m,B}^{(n)} \end{bmatrix} \in \mathbb{R}^{B \times T}. \tag{15}$$

Then, we can define a dataset type two $\mathcal{D}_{T_2}$ as

$$\mathcal{D}_{T_2} = \left\{ \left( V_m^{(n)}, y^{(n)} \right) : m = [\![M]\!] \right\}_{n=1}^{N}. \tag{16}$$

Similarly, in the literature they go further and create datasets with tensors. To achieve this from a dataset type two, we stack the $V_m$ matrices in a tensor obtaining $I^{(n)} = \mathbb{R}^{M \times B \times T}$. This results in a dataset type three

$$\mathcal{D}_{T_3} = \left\{ \left( I^{(n)}, y^{(n)} \right) \right\}_{n=1}^{N}. \tag{17}$$

From the last part, some important points and questions are: (1) What are the advantages or disadvantages of using $\mathcal{D}_{T_1}$, $\mathcal{D}_{T_2}$, or $\mathcal{D}_{T_3}$? (if any). (2) Do different dataset types have the same vulnerabilities? (3) Can we quantify the difference between dataset types by using Optimal Transport?

## 5 EXPERIMENTAL RESULTS

To explore the advantages and disadvantages of using different dataset types, we carry trained different classifiers with diverse dataset types and models. In specific, we use the MNIST and Fashion-MNIST datasets. Although these datasets are not from a different domain (e.g., CPSs), they are helpful to assess what are the implications of using them with other formats (i.e., $\mathcal{D}_{T_1}$ or $\mathcal{D}_{T_2}$) To assess the advantages, we measure the accuracy of the test regular data. To evaluate the disadvantages, we also measure the accuracy but in perturbed test data. The perturbations are done with the fast gradient sign method (Goodfellow et al., 2014). All datasets consists in examples $x \in \mathbb{R}^{28 \times 28}$. According to our previous classifications, they are dataset type 2, $\mathcal{D}_{T_2}^{\text{MNIST}}$ and $\mathcal{D}_{T_2}^{\text{Fashion}}$, respectively.

These datasets can be transformed into datasets type 1 by stacking the columns into a vector resulting in examples $x \in \mathbb{R}^{784}$. This would result in datasets type 1, $\mathcal{D}_{T_1}^{\text{MNIST}}$ and $\mathcal{D}_{T_1}^{\text{Fashion}}$, respectively. We create classifiers for both dataset types $\mathcal{D}_{T_1}$ and $\mathcal{D}_{T_2}$ based on fully connected neural networks (Dense) and convolutional neural networks (Conv), respectively. The fully connected neural network (Dense) has two dense layers of size 784 and 128, resulting in $101,770$ parameters. We use a CNN consisting of two convolutional layers with 32 and 62 filters, respectively, each followed by 2×2 max-pooling, and a fully connected layer of size 256 resulting in $84,553$ parameters (Conv1). Also, to see that in the CNN model, the number of parameters does not affect the classifier output, we use a second CNN (Conv2) with two fully connected layers of size 256 and 128 resulting in $116,168$ parameters.

We train these classifiers in two ways: (1) we train them on regular data, and (2) we also adversary train the networks (AT or Adv) with regular and adversarial samples, with the method proposed by Madry et al. (2017).

### 5.1 ACCURACY AND DISTANCE BETWEEN ORIGINAL AND ADVERSARIAL EXAMPLES

The results for the MNIST-Fashion dataset are shown in Table 1 (with $\epsilon = 0.05$). We perturbed both datasets $\mathcal{D}_{T_1}$ and $\mathcal{D}_{T_2}$ with the fast gradient descent method (FGDM) (Goodfellow et al., 2014). As result, we obtain adversarial or perturbed datsets that $\mathcal{D}_{T_1}^{\text{P}}$ and $\mathcal{D}_{T_2}^{\text{P}}$. Then, we measure the distance between the regular and perturbed datasets with the optimal transport framework. The results are

$$\text{OT}_\epsilon(\mathcal{D}_{T_1}, \mathcal{D}_{T_1}^{\text{P}}) = 1.05, \ \text{OT}_\epsilon(\mathcal{D}_{T_2}, \mathcal{D}_{T_2}^{\text{P}}) = 0.84. \tag{18}$$

In this case

$$\text{OT}_\epsilon(\mathcal{D}_{T_1}, \mathcal{D}_{T_1}^\text{P}) > \text{OT}_\epsilon(\mathcal{D}_{T_2}, \mathcal{D}_{T_2}^\text{P}). \tag{19}$$

The effect of this is that adversarial examples look more similar for $\mathcal{D}_{T_2}$. We hyphotesize that this is correlated with the classifier accuracy on the perturbed dataset. If the distance between the regular and perturbed dataset is larger, then the accuracy on the adversarial trained model will be better as shown in Table 1. We repeat the same experiment for different $\epsilon$ ranging from $0$ to $0.3$ with increments of $0.05$. The results are shown in Fig. 2. In this Figure, we make two observations. (1) The accuracy for the adversary trained model with dataset type 1 for the perturbed dataset is higher than its counterpart with dataset type 2. (2) The OT distance between the regular and perturbed datasets type 1 is bigger than the distance for dataset type 2.

The results for the MNIST dataset (with $\epsilon = 0.05$) are shown in Table 2. The OT distances between regular and perturbed datasets are

$$\text{OT}_\epsilon(\mathcal{D}_{T_1}, \mathcal{D}_{T_1}^\text{P}) = 16.24, \ \text{OT}_\epsilon(\mathcal{D}_{T_2}, \mathcal{D}_{T_2}^\text{P}) = 9.64. \tag{20}$$

We can see a similar effect, the distance for the dataset type 1 is larger than the distance for the dataset type 1. This results in bigger increase of accuracy in the adversary train classifier as shown in Table 2. We repeat the same experiment for different $\epsilon$ ranging from $0$ to $0.3$ with increments of $0.05$. The results are shown in Fig. 3. In this Figure, we make two observations. (1) The accuracy for the adversary trained model with dataset type 1 for the perturbed dataset is higher than its counterpart with dataset type 2. (2) The OT distance between the regular and perturbed datasets type 1 is bigger than the distance for dataset type 2.

This shows that we must be aware when selecting the dataset transformation or ML model architecture across different domains. While using datasets type 2 improves the accuracy, it also makes models more sensible to adversarial examples.

| Model | Accuracy (%) | | | | No. of parameters |
| --- | --- | --- | --- | --- | --- |
| | Regular | Perturbed | Regular (AT) | Perturbed (AT) | |
| Conv1 ($\mathcal{D}_{T_2}$) | 90 | 39 | 88 | 84 | 84,552 |
| Conv2 ($\mathcal{D}_{T_2}$) | 90 | 44 | 89 | 85 | 116,168 |
| Linear ($\mathcal{D}_{T_1}$) | 86 | 59 | 87 | 97 | 101,770 |

Table 1: Model comparison for MNIST-Fashion dataset with $\epsilon = 0.05$.

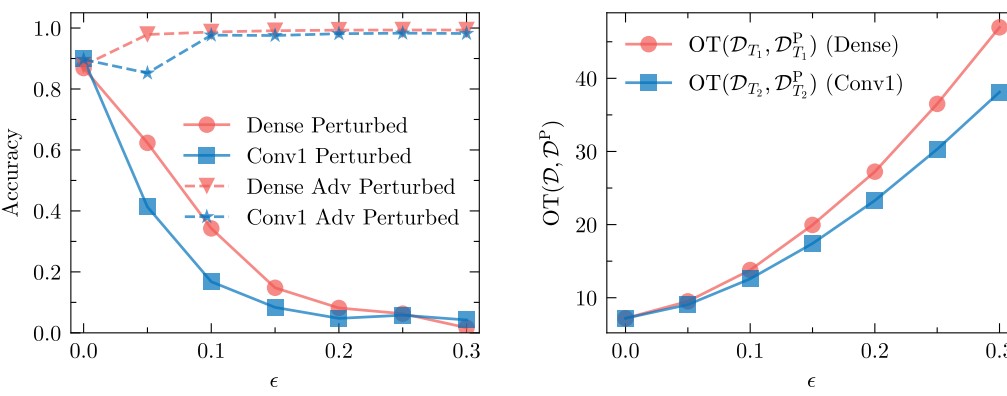

(a) Accuracy for perturbed examples for the regular and adversarial trained classifiers.

(b) OT distance between the regular and perturbed test set.

Figure 2: Accuracy and OT distance for the MNIST-Fashion dataset.

## 5.2 COMPARISON OF DISTANCES BETWEEN SAMPLES FOR DIFFERENT MODELS

To further investigate the effect of adversarial examples for different models, we compute the distances among the original and pertubed datasets for both the FCN and the CNN. Suppose we have a

| Model | Accuracy (%) | | | | No. of parameters |
|---|---|---|---|---|---|
| | Regular | Perturbed | Regular (AT) | Perturbed (AT) | |
| Conv1 ($\mathcal{D}_{T_2}$) | 99 | 47 | 99 | 99.22 | 84,552 |
| Conv2 ($\mathcal{D}_{T_2}$) | 99 | 65 | 99 | 98 | 116,168 |
| Linear ($\mathcal{D}_{T_1}$) | 98 | 5 | 98 | 99.88 | 101,770 |

Table 2: Model comparison for dataset 2.

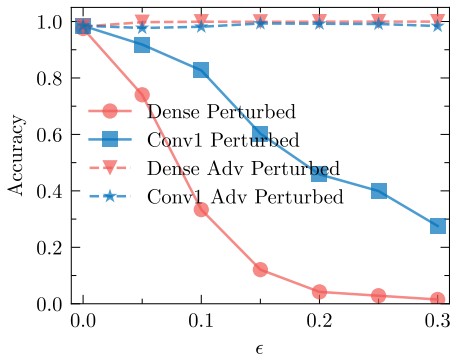

(a) Accuracy for perturbed examples for the regular and adversarial trained classifiers.

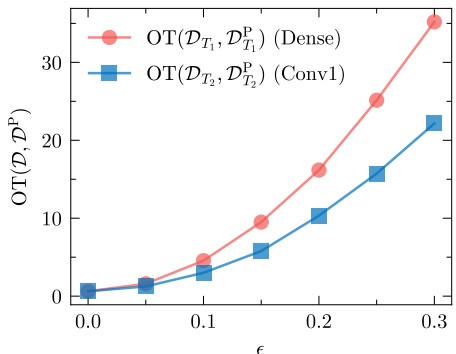

(b) OT distance between the regular and perturbed test set.

Figure 3: Accuracy and OT distance for the MNIST dataset.

dataset with $N$ samples $\mathcal{D} = \{x_n\}_{n=1}^{N}$. Then, we can compute the distance of an example with all the remaining dataset with the OT framework. Mathematically, it is expressed as follows

$$\text{emd}_i = \text{OT}_i \left( x_i, \{x_n \in \mathcal{D} : x_n \neq x_i\} \right), \quad \text{for } i = 1, \ldots, N. \tag{21}$$

With the computed distances for every sample, we define the set as $\text{EMD} = \{\text{emd}_n\}_{n=1}^{N}$. We can interpret this as an empirical PDF of distances. The results for the MNIST dataset for the FCN and the CNN are shown in Fig. 4. We can see in Fig. 4b that the distances between the original and perturbed examples barely move. Fig. 4a, on the other hand, shows that interestingly the perturbed distances are smaller than the original ones. This could be the reason why the CNN is more vulnerable to adversarial attacks than the FCN. In other words, the FCN is more resilient to adversarial attacks because the distribution of distances between the original and perturbed examples is not entirely overlapping as in the case of the CNN case. We can see similar results for the MNIST-Fasion dataset in in Fig. 5.

## 6 CONCLUSION

This paper showed that using different dataset types and neural network architectures has benefits and downsides. When adapting ML techniques across domains, practitioners must be aware of the trade-offs on selecting a given dataset type format or model architecture. While using a convolutional neural network improves the model's accuracy, making it more vulnerable to adversarial examples. We showed that it is possible to infer how vulnerable is a given model with respect to another one by computing their corresponding OT distances between the regular and perturbed datasets. It was proven that a lower OT would result in a more vulnerable model against adversarial examples.

## REFERENCES

Rasim Alguliyev, Yadigar Imamverdiyev, and Lyudmila Sukhostat. Cyber-physical systems and their security issues. *Computers in Industry*, 100:212–223, 2018.

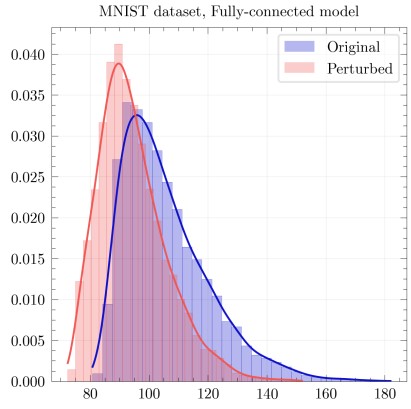

(a) Distance's PDF with a FCN model.

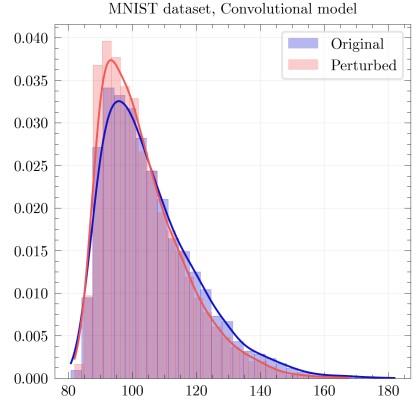

(b) Distance's PDF with a CNN model.

Figure 4: PDFs of MNIST dataset distances.

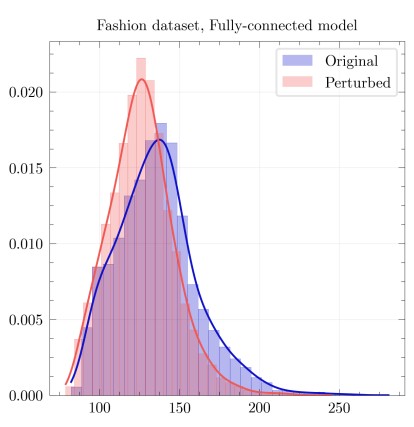

(a) Distance's PDF with a FCN model.

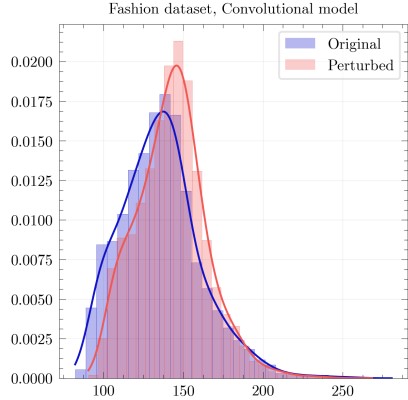

(b) Distance's PDF with a CNN model.

Figure 5: PDFs of Fashion dataset distances.

Oyeniyi Akeem Alimi, Khmaies Ouahada, and Adnan M Abu-Mahfouz. A review of machine learning approaches to power system security and stability. *IEEE Access*, 8:113512–113531, 2020.

Jason Altschuler, Jonathan Niles-Weed, and Philippe Rigollet. Near-linear time approximation algorithms for optimal transport via sinkhorn iteration. In I Guyon, U V Luxburg, S Bengio, H Wallach, R Fergus, S Vishwanathan, and R Garnett (eds.), *Advances in Neural Information Processing Systems 30*, pp. 1964–1974. Curran Associates, Inc., 2017.

David Alvarez-Melis and Nicolo Fusi. Geometric dataset distances via optimal transport. *arXiv preprint arXiv:2002.02923*, 2020.

Anish Athalye, Logan Engstrom, Andrew Ilyas, and Kevin Kwok. Synthesizing robust adversarial examples. *arXiv preprint arXiv:1707.07397*, 2017.

Victoria A Banks, Katherine L Plant, and Neville A Stanton. Driver error or designer error: Using the perceptual cycle model to explore the circumstances surrounding the fatal tesla crash on 7th may 2016. *Safety science*, 108:278–285, 2018.

Huaxiang Cai and David J Hill. A real-time continuous monitoring system for long-term voltage stability with sliding 3d convolutional neural network. *International Journal of Electrical Power & Energy Systems*, 134:107378, 2022.

Nicholas Carlini and David Wagner. Defensive distillation is not robust to adversarial examples. *arXiv preprint arXiv:1607.04311*, 2016.

Nicholas Carlini and David Wagner. Adversarial examples are not easily detected: Bypassing ten detection methods. In *Proceedings of the ACM Workshop on Artificial Intelligence and Security*, pp. 3–14, 2017.

Yize Chen, Yishen Wang, Daniel Kirschen, and Baosen Zhang. Model-free renewable scenario generation using generative adversarial networks. *IEEE Transactions on Power Systems*, 33(3): 3265–3275, 2018.

Marco Cuturi. Sinkhorn distances: Lightspeed computation of optimal transport. In C J C Burges, L Bottou, M Welling, Z Ghahramani, and K Q Weinberger (eds.), *Advances in Neural Information Processing Systems 26*, pp. 2292–2300. Curran Associates, Inc., 2013.

Murat Dikmen and Catherine Burns. Trust in autonomous vehicles: The case of tesla autopilot and summon. In *International Conference on Systems, Man, and Cybernetics*, pp. 1093–1098. IEEE, 2017.

Laurine Duchesne, Efthymios Karangelos, and Louis Wehenkel. Recent developments in machine learning for energy systems reliability management. *Proceedings of the IEEE*, 108(9):1656–1676, 2020.

Aude Genevay, Léna\"ic Chizat, Francis Bach, Marco Cuturi, and Gabriel Peyré. Sample complexity of sinkhorn divergences. In Kamalika Chaudhuri and Masashi Sugiyama (eds.), *Proceedings of Machine Learning Research*, volume 89 of *Proceedings of Machine Learning Research*, pp. 1574–1583. PMLR, 2019.

Ian J Goodfellow, Jonathon Shlens, and Christian Szegedy. Explaining and harnessing adversarial examples. *arXiv preprint arXiv:1412.6572*, 2014.

Ankita Gupta, Gurunath Gurrala, and PS Sastry. An online power system stability monitoring system using convolutional neural networks. *IEEE Transactions on Power Systems*, 34(2):864–872, 2018.

Nasser Jazdi. Cyber physical systems in the context of industry 4.0. In *International conference on automation, quality and testing, robotics*, pp. 1–4. IEEE, 2014.

Il-Young Jeong, Subin Lee, Yoonchang Han, and Kyogu Lee. Audio event detection using multiple-input convolutional neural network. *Detection and Classification of Acoustic Scenes and Events (DCASE)*, 2017.

L Kantorovitch. On the translocation of masses. *Dokl. Akad. Nauk SSSR*, 37(7-8):227–229, 1942. ISSN 0002-3264.

Alexey Kurakin, Ian Goodfellow, Samy Bengio, Yinpeng Dong, Fangzhou Liao, Ming Liang, Tianyu Pang, Jun Zhu, Xiaolin Hu, Cihang Xie, et al. Adversarial attacks and defences competition. In *The NIPS'17 Competition: Building Intelligent Systems*, pp. 195–231. Springer, 2018.

Aleksander Madry, Aleksandar Makelov, Ludwig Schmidt, Dimitris Tsipras, and Adrian Vladu. Towards deep learning models resistant to adversarial attacks. *arXiv preprint arXiv:1706.06083*, 2017.

Antoine Marot, Sami Tazi, Benjamin Donnot, and Patrick Panciatici. Guided machine learning for power grid segmentation. In *2018 IEEE PES Innovative Smart Grid Technologies Conference Europe (ISGT-Europe)*, pp. 1–6. IEEE, 2018.

Mostafa Mohammadpourfard, Ashkan Sami, and Yang Weng. Identification of false data injection attacks with considering the impact of wind generation and topology reconfigurations. *IEEE Transactions on Sustainable Energy*, 9(3):1349–1364, 2017.

Mostafa Mohammadpourfard, Yang Weng, and Mohsen Tajdinian. Benchmark of machine learning algorithms on capturing future distribution network anomalies. *IET Generation, Transmission & Distribution*, 13(8):1441–1455, 2019.

Seyed-Mohsen Moosavi-Dezfooli, Alhussein Fawzi, Omar Fawzi, and Pascal Frossard. Universal adversarial perturbations. In *Proceedings of the IEEE conference on computer vision and pattern recognition*, pp. 1765–1773, 2017.

Felix O Olowononi, Danda B Rawat, and Chunmei Liu. Resilient machine learning for networked cyber physical systems: A survey for machine learning security to securing machine learning for cps. *IEEE Communications Surveys & Tutorials*, 23(1):524–552, 2020.

Gabriel Peyré and Marco Cuturi. Computational optimal transport. *Foundations and Trends® in Machine Learning*, 11(5-6):355–607, 2019. ISSN 1935-8237. doi: 10.1561/2200000073.

Huiying Ren, Z Jason Hou, Bharat Vyakaranam, Heng Wang, and Pavel Etingov. Power system event classification and localization using a convolutional neural network. *Frontiers in Energy Research*, 8:327, 2020.

Yossi Rubner, Carlo Tomasi, and Leonidas J Guibas. The earth mover's distance as a metric for image retrieval. *Int. J. Comput. Vis.*, 40(2):99–121, November 2000. ISSN 0920-5691, 1573-1405. doi: 10.1023/A:1026543900054.

Zhongtuo Shi, Wei Yao, Lingkang Zeng, Jianfeng Wen, Jiakun Fang, Xiaomeng Ai, and Jinyu Wen. Convolutional neural network-based power system transient stability assessment and instability mode prediction. *Applied Energy*, 263:114586, 2020.

Christian Szegedy, Wojciech Zaremba, Ilya Sutskever, Joan Bruna, Dumitru Erhan, Ian Goodfellow, and Rob Fergus. Intriguing properties of neural networks. *arXiv preprint arXiv:1312.6199*, 2013.

Cédric Villani. *Topics in Optimal Transportation*. American Mathematical Soc., 2003. ISBN 9780821833124.

Cédric Villani. *Optimal transport, Old and New*, volume 338. Springer Science & Business Media, 2008. ISBN 9783540710493.

Bo Wang, Biwu Fang, Yajun Wang, Hesen Liu, and Yilu Liu. Power system transient stability assessment based on big data and the core vector machine. *IEEE Transactions on Smart Grid*, 7 (5):2561–2570, 2016.

Huaizhi Wang, Haiyan Yi, Jianchun Peng, Guibin Wang, Yitao Liu, Hui Jiang, and Wenxin Liu. Deterministic and probabilistic forecasting of photovoltaic power based on deep convolutional neural network. *Energy conversion and management*, 153:409–422, 2017.

Daoqiang Xu, Zhixin Li, Shihai Yang, Zigang Lu, Haowei Zhang, Wenguang Chen, and Qingshan Xu. A classified identification deep-belief network for predicting electric-power load. In *2018 2nd IEEE Conference on Energy Internet and Energy System Integration (EI2)*, pp. 1–6. IEEE, 2018a.

Yong Xu, Qiuqiang Kong, Qiang Huang, Wenwu Wang, and Mark D Plumbley. Convolutional gated recurrent neural network incorporating spatial features for audio tagging. In *2017 International Joint Conference on Neural Networks (IJCNN)*, pp. 3461–3466. IEEE, 2017.

Yong Xu, Qiuqiang Kong, Wenwu Wang, and Mark D Plumbley. Large-scale weakly supervised audio classification using gated convolutional neural network. In *2018 IEEE international conference on acoustics, speech and signal processing (ICASSP)*, pp. 121–125. IEEE, 2018b.

