# OpenReview forum: "Dataset transformations trade-offs to adapt machine learning methods across domains"
_ICLR.cc/2022/Conference — ICLR 2022 Submitted_

### Official Review · Reviewer_JxT2 · 2021-10-30

**Correctness:** 3
**Technical Novelty And Significance:** 1
**Empirical Novelty And Significance:** 2
**Recommendation:** 3
**Confidence:** 5

**Details Of Ethics Concerns:**

I do not find ethics concerns.

**Main Review:**

Pros:

- In general, the authors clearly deliver their ideas and experimental procedure.

- While people are introducing ML methods to more and more real-world problems, the topic covered in this paper is certainly becoming more attractive.

Cons and feedbacks:

- The problem is not well formulated. The idea of studying different input formates could be helpful when solving practical problems, but I would suggest the authors formulate this problem by starting from even simple NN models and assumptions on datasets. Is it possible to use an objective to describe this discrepancy caused by data transformations?

- Lack experiments. The authors are inspired by ML problems on CPSs datasets, however, the only experiment in this paper is on the MNIST and Fashion-MNSIT datasets, and the experiment on these part (MNIST, fashionMNIST) are already provided by public codebases [0]. Experiments on datasets from power systems, robotics, or autonomous driving are expected. There are multiple public datasets from CPSs mentioned above [1, 2, 3]

- The definition of “domain” and “adapt”. This paper defines the domain as a broad area of applications such as CV, audio recognition, and others, and the term “adapt” means changing the data formate. However, in many existing studies, the term domain is usually associated with the distribution of data [4]. Consider an autonomous driving case where we collect two datasets (image and vehicle trajectory) for each weather (sunny and raining) and thus have four datasets in total, how should we divide these four datasets into different domains?

- The usage of the Optimal Transport Data Distance. While this is a promising prior study on dataset geometric distances, one assumption is that the label-feature distributions are Gaussian. Does that still hold for the CPSs dataset that may contain continuous signals? I would like to hear more from the authors on this point.

- It is a good idea to use adversarial attacks to evaluate the transformations. It would be interesting to utilize the “Wasserstein adversarial attack” [5] and see if this method correlates with the optimal transport data distance.


[0]: Link: github.com/microsoft/otdd

[1]: Link: www.kaggle.com/robikscube/hourly-energy-consumption

[2]: Geiger, Andreas, et al. "Vision meets robotics: The kitti dataset." The International Journal of Robotics Research 32.11 (2013): 1231-1237.

[3]: Mandlekar, Ajay, et al. "Roboturk: A crowdsourcing platform for robotic skill learning through imitation." Conference on Robot Learning. PMLR, 2018.

[4]: Courty, Nicolas, et al. "Optimal transport for domain adaptation." IEEE transactions on pattern analysis and machine intelligence 39.9 (2016): 1853-1865.

[5]: Wong, Eric, Frank Schmidt, and Zico Kolter. "Wasserstein adversarial examples via projected Sinkhorn iterations." International Conference on Machine Learning. PMLR, 2019.

**Summary Of The Paper:**

This paper aims to study the effects of different transformation methods when processing a dataset for supervised deep learning tasks. The authors conducted one experiment to show that different data arranging methods leads to different vulnerability for adversarial attacks. In the experimental procedure, an optimal transport based method is utilized to measure the distance between a dataset and its perturbed version.

**Summary Of The Review:**

In general, this paper raised an interesting problem and studied the obstacle when people want to apply ML methods to CPSs data. The authors conducted experiments with existing methods on OT, deep learning, and adversarial attack. However, I feel the results and analysis are still in their preliminary form. This paper could meet the acceptance requirements with a clear problem formulation, more extensive experimental results with CPSs data, and theoretical analysis.

---

### Official Review · Reviewer_57HV · 2021-11-03

**Correctness:** 2
**Technical Novelty And Significance:** 1
**Empirical Novelty And Significance:** 1
**Recommendation:** 3
**Confidence:** 4

**Main Review:**

- The draft fails to support the claims strongly. For instance, they do not consider data from CPS, instead they consider well known datasets from computer vision on which various DNN architectures have already been explored. Instead, authors should have worked with a different domain as claimed.

- Also, the datasets MNIST and Fashion MNIST are similar in some of the important aspects: both gray scale, same number of classes, and number of samples (complexity), etc. This does not support the draft's claim that it works with diverse dataset types.

- Experimental analysis presented in the paper is very weak. Only a couple simplest datasets are considered to train simple CNN and MLP classifier architectures. This kind of analysis is not very convincing to generalize from.

- It is not very clear why authors consider adversarial robustness to investigate the disadvantage of the dataset conversion.

- The take-away (according to the abstract) is that  the data conversion is not always needed or beneficial. This is not very informative or novel, since it is understandable that the data formats need to be understood and domain knowledge also is required to avoid any structure or information loss while processing,

**Summary Of The Paper:**

- Draft attempts to understand the effect of converting dataset format while applying an ML technique. It argues that converting the target data into a format on which the specific ML technique was shown to be successful is suboptimal. Via simple experiments it shows that the data sample format conversion need not be advantageous.

**Summary Of The Review:**

- Draft is incomplete in multiple aspects. It promises much in the abstract (or goals) but fails to deliver it. It is supposed to be a comprehensive empirical analysis, but the provided experiments fall short of it.

---

### Official Review · Reviewer_GQQh · 2021-11-07

**Correctness:** 2
**Technical Novelty And Significance:** 1
**Empirical Novelty And Significance:** 1
**Recommendation:** 1
**Confidence:** 4

**Main Review:**

The authors find that different representations and models give different performance, which is not surprising. This is a known fact - in practice, domain specific models are generally trained tailored to the task at hand. These models encode implicit biases needed for the task. For vision tasks, convnets are trained and for NLP, transformers are used. Researchers are aware of this, and the conclusions of this paper add no value in my opinion. Furthermore, the experiments are performed only on MNIST with minimal architectural choices, and the results are not conclusive in any form. I recommend strongly rejecting the paper.

**Summary Of The Paper:**

This paper attempts to study the effect of different transformations of input datasets, and the performance of the resulting models thereof. In particular, authors consider time-series data which is then represented in (1) time-series format, (2) vectorized form, (3) tensorized form. For each of the representation, a different neural network is trained and the performance is compared. Experiments are shown on MNIST dataset.

**Summary Of The Review:**

The conclusions of the paper add no value, and the experiments are not performed well.

---

### Decision · Program_Chairs · 2022-01-20

**Decision:**

Reject

**Comment:**

Unfortunately, reviewers unanimously agreed that this paper does not meet the ICLR acceptance standards, citing generally unpolished experiments. I would recommend substantially expanding the experimental results in the future.